# Enhanced Bandwidth Broadening of Infrared Reflector Based on Polymer Stabilized Cholesteric Liquid Crystals with Poly(N-vinylcarbazole) Used as Alignment Layer

**DOI:** 10.3390/polym13142238

**Published:** 2021-07-08

**Authors:** Limin Zhang, Qiumei Nie, Xiao-Fang Jiang, Wei Zhao, Xiaowen Hu, Lingling Shui, Guofu Zhou

**Affiliations:** 1SCNU-TUE Joint Lab of Device Integrated Responsive Materials (DIRM), National Center for International Research on Green Optoelectronics, South China Normal University, Guangzhou 510006, China; 2019023282@m.scnu.edu.cn (L.Z.); shenyouhhh@163.com (Q.N.); weizhao@m.scnu.edu.cn (W.Z.); guofu.zhou@m.scnu.edu.cn (G.Z.); 2Guangdong Provincial Key Laboratory of Optical Information Materials and Technology, Institute of Electronic Paper Displays, South China Academy of Advanced Optoelectronics, South China Normal University, Guangzhou 510006, China; shuill@m.scnu.edu.cn; 3Laboratory of Quantum Engineering and Quantum Material, School of Physics and Telecommunication Engineering, South China Normal University, Guangzhou 510006, China; jiangxf@scnu.edu.cn

**Keywords:** cholesteric liquid crystals, alignment layer, reflection bandwidth, infrared reflectors

## Abstract

Alignment layer plays a critical role on liquid crystal (LC) conformation for most LC devices. Normally, polyimide (PI) or polyvinyl alcohol (PVA), characterized by their outstanding thermal and electrical properties, have been widely applied as the alignment layer to align LC molecules. Here, we used a semi-conductive material poly(N-vinylcarbazole) (PVK) as the alignment layer to fabricate the cholesteric liquid crystal (CLC) device and the polymer-stabilized cholesteric liquid crystals (PSCLC)-based infrared (IR) reflectors. In the presence of ultraviolet (UV) irradiation, there are hole–electron pairs generated in the PVK layer, which neutralizes the impurity electrons in the LC–PVK junction, resulting in the reduction in the built-in electric field in the LC device. Therefore, the operational voltage of the CLC device switching from cholesteric texture to focal conic texture decreases from 45 V to 30 V. For the PSCLC-based IR reflectors with the PVK alignment layer, at the same applied electric field, the reflection bandwidth is enhanced from 647 to 821 nm, ranging from 685 to 1506 nm in the IR region, which makes it attractive for saving energy as a smart window.

## 1. Introduction

Smart windows that enable electronic control of visible and infrared light have attracted extensive attention in both academia and industry due to their potential application for architectural aesthetics, occupant comfort and energy efficiency [1,2]. In buildings, energy savings of 20% can be made through smart windows due to reduced lighting, heating and cooling costs [2]. Currently, various technologies are known to fabricate such windows including electrochromic (EC) [3,4,5], suspended particle electrophoresis [6,7], polymer dispersed liquid crystals (PDLC) [8,9,10], polymer stabilized liquid crystals (PSLC) [11,12,13,14,15,16] and polymer stabilized cholesteric liquid crystals (PSCLC) [17,18,19,20,21,22]. Among them, electrically tunable infrared (IR) reflecting/transmitting windows based on PSCLC, which can selectively reflect different fractions of infrared radiation on demand without affecting the transmittance in the visible region, appear to be very attractive [21,22].

Cholesteric liquid crystals (CLCs), composed of nematic liquid (LC) and chiral molecule that generates a helical twist between the LC layers, allow the selective reflection of circularly polarized incident light of the same chirality with the certain reflection bandwidth of the helix. The wavelength of the reflected light is determined by the helical pitch of CLC (p). The bandwidth of the reflected light is defined as $\Delta \lambda =\Delta n\times p=\left(n_{e}-n_{0}\right)\times p$, where Δn is the birefringence index of the host LC, *n_e_* and *n*_0_ are the extraordinary refractive index and ordinary refractive index, respectively [23,24]. The reflection bandwidth (Δλ) of CLC is typically 50–100 nm in the visible/infrared region. However, more than 75% of the energy of IR light from the sun lies between 700 and 1400 nm, which is responsible for most of the heating up of buildings and automobiles [17]. Therefore, it is necessary to fit the reflection bandwidth of IR reflector into the above wavelength range for effectively resisting heat from the solar irradiation.

Many methods were reported to realize the reflection bandwidth broadening of CLC. Broer et al. have obtained bandwidth broadening in a CLC film doped with an ultraviolet (UV) light absorbing dye, which was used to create an intensity gradient through the thickness of a solid cholesteric polymer network, resulting in a pitch gradient through the cell thickness [25]. Yang et al. report a method that blended different polymerizable monomers and dye into the PSCLC system, and the broadband reflection wavelength was significantly increased to 550–2350 nm after polymerization [26]. However, the bandwidth through the above methods was static once frozen by the polymerization process, which cannot be tunable. Electric-induced bandwidth broadening of PSCLC-based IR reflector is attractive due to their reversible dynamic notch broadening. T. J. White’s group has done much pioneer research on electrically induced bandwidth broadening [27,28,29]. They demonstrated that the electric-induced changes to the reflection bandwidth was ionic in nature and attributed to the pitch distortion caused by the displacement of the polymer network. Recent research reveals that both the preparation conditions and constituent components, including cell gap, photo-initiator concentration, and chemical structure of the monomers and the chiral dopant, have an impact on the electrically induced reflection bandwidth broadening and dynamic response of the IR reflectors [19,22,30,31]. Khandelwal adopted a long and flexible ethylene glycol crosslinker twin molecule to improve the ability of the polymer stabilizing network to trap the cation impurities, which resulted in a nine-fold enhancement in reflection bandwidth [21]. Yang et al. developed a chiral nematic liquid crystal doped with chiral ionic liquid composite where the reflection bandwidth can be controllable; the composite can also be changed into a light scattering state by an electric field [32]. Rather than using complicated constituent components, in this work, we demonstrated enhanced bandwidth broadening of PSCLC devices by using poly(N-vinylcarbazole) (PVK) as the alignment layer, which can form a LC–PVK junction and reduce the built-in electric field in the device [33], consequently lowering the driving voltage of the device and enhanced bandwidth broadening.

PVK is an organic semiconductive polymer for hole transport, which has been widely applied as an electro-optic material for its photoconductive characteristics [34,35]. In the meanwhile, PVK has been applied previously for fabricating broadband CLC devices through particular thermally induced phase separation (TIPS) [36]. The method involves a combination of the dissolution process of PVK dissolved in the LC layer, and it has been successfully applied for fabricating scattering mode LC light shutters with low driving voltage and fast response [37]. In our work, we replaced traditional alignment layers, polyimide (PI) or polyvinyl alcohol (PVA), with PVK to fabricate CLC and PSCLC devices, instead of diffusing PVK into LC layer by TIPS. It was found that, when using PVK as the alignment layer, the voltage of the CLC device switching from the cholesteric phase to the focal conic phase decreased from 45 V to 30 V. For the PSCLC-based IR reflectors, the reflection bandwidth was enhanced from 647 to 821 nm at same applied voltage. Our results demonstrate that the PVK layer can be used as the alignment layer to achieve CLC devices with low operational voltage and enhanced bandwidth broadening.

## 2. Materials and Methods

Two mixtures were prepared. Mixture 1 consisted of nematic non-reactive LC HTW138200–100 (Δε = 16.5, purchased from HCCH, Nanjing, China) mixed with left-handed chiral dopant S1011 (purchased from HCCH, Nanjing, China). Mixture 2 consisted of both reactive monomer HCM009 (purchased from HCCH, Nanjing, China) and nematic non-reactive LC HNG-30400–200 (Δε = −8.3, purchased from HCCH, Nanjing, China) mixed with a photo-initiator Irgacure-651 (purchased from HEOWNS, Tianjin, China) and left-handed chiral dopant S811 (purchased from Merck, Darmstadt, Germany). Table 1 shows the concentration (by weight) of components in the mixtures. Mixture 1 was prepared by mixing together 91.1% HTW138200–100 and 8.9% of the chiral dopant S1011. Mixture 2 was prepared by mixing together 82.84 wt % of the nematic HNG-30400–200, 5 wt % of the diacrylate monomer HCM009, 11.16 wt % of the chiral dopant S811 and 1 wt % of the photo-initiator Irgacure-651. Scheme 1 shows the molecular structures of the chemicals. To explore the application of reflecting IR light, the concentration of the chiral dopant in mixture 2 was selected to make the reflection band of the PSCLCs cells centered near 1100 nm. Powdered PVK was dissolved in chlorobenzene at a weight ratio of 4%. The solution was then spin-coated onto a cleaned ITO-coated glass substrate. The substrate was pre-baked at 80 °C for 20 min followed by being post-baked at 120 °C for 2 h. The alignment film on the opposite substrate was PVA. The PVK- and PVA-coated glass slides were assembled in an anti-parallel way to an empty cell by sandwiching them with a 20 μm pearl powder spacer. Moreover, two PVA-coated glass substrates and two PVK-coated glass substrates were combined to fabricate empty cells as control devices.

Sample cells and LC mixtures were heated to temperatures around 60 °C before filling. Cell filling was fulfilled by capillary action between the fluidic LCs and micro-scale cell gaps. For mixture 2, after filling, the cells were exposed to UV light (33 mW cm^−2^) to polymerize.

## 3. Results and Discussion

For most LC devices, the alignment layer plays a critical role on LC conformation. Here, the orientation effect of PVK was firstly investigated by polarized microscopy (POM). The orientation effect of traditional alignment layer PVA was also investigated for comparison. Nematic liquid crystal (5CB) was filled into the cells constructed with three different alignment layer combinations, PVA–PVA, PVA–PVK and PVK–PVK. Figure 1 shows the POM images of the LC cells. For the cells with PVA–PVA and PVA–PVK alignment layers, a bright and dark field appeared alternately and the rotation angle of the cell increased from 0 to 45°, 90°, 135°and 180° (Figure 1a,b), which suggests a homogeneous parallel orientation. For the cell with the PVK–PVK alignment layer, the alternating bright and dark field could also be observed (Figure 1c). However, there were some unexpected lines of disinclinations that appeared (Figure 1c), which probably influenced the orientation of the LC molecules in the cell. In order to prove this, we further calculated the order parameter (S) of a 5CB doped with a dichroic dye (RL-002, purchased from HCCH) in the cells with these three alignment layers.

Figure 2 presents the absorption of the dichroic dye in the nematic cells with different alignment layers. The absorption of incident light that is polarized parallel and perpendicular to the principal axes of the dichroic dye is maximal and minimal, respectively. In the PVA–PVA cell, the parallel absorption of the dye is stronger than that in the PVA–PVK cell; it is weakest in the PVK–PVK cell, which suggests that the PVA–PVA cell has the best parallel orientation effect while PVK–PVK performs the worst. From the absorption, the order parameter can be calculated according to the equation as follows [38]:(1)$$S=\frac{2\left(A_{∥}-A_{⊥}\right)}{\left(A_{∥}+2A_{⊥}\right)\left(3\cos^{2}\gamma -1\right)}$$
where $A_{∥}$ and $A_{⊥}\;$ are the absorbances for light polarized parallel or perpendicular to the principal axes of the dye, respectively, γ is the angle between the principal axes and transition dipole moment of the LC. The calculated order parameter for a dye concentration of 0.5 wt % in 5CB with different alignment layers is summarized in Table 2. The LC cell with the PVA–PVK alignment layer has an order parameter of 0.53, which is close to that of LC cell with the PVA–PVA alignment layer (0.57). The LC cell with PVK–PVK alignment layer has the lowest order parameter of 0.35. This can be attributed to an unordered alignment of PVK–PVK cell, which results in a weak dye alignment effect by the host LC and weak interactions between the anisotropic LC and the dye molecules [39]. From the above results, the aligning effect of the PVA–PVK combination is comparable to that of the PVA–PVA combination. Therefore, the PVA–PVA and the PVA–PKV alignment layer are used for our further investigation.

In order to check the effect of the PVK alignment layer on reducing driving voltage of LC devices, we firstly fabricated CLC devices with PVA–PVK and PVA–PVA alignment layers, respectively. Mixture 1 was injected into anti-parallel rubbed PVA–PVA cell and PVA–PVK cell, respectively. Figure 3 illustrates the switching behavior of the CLC cells upon application of voltage. For both the PVA–PVK cell and the PVA–PVA cell, there was a reflection band of the transmittance spectra in initial state (0 V) because of cholesteric texture (Figure 3a,b). After applying voltage, for the PVA–PVA cell, the transmittance almost remained the same until the voltage increased to 45 V, at which point the transmittance dramatically dropped to about 15% over the whole visible light, and the reflection band nearly disappeared (Figure 3a), corresponding to a focal conic texture. For the PVA–PVK cell, however, the voltage for switching the CLC from the cholesteric texture to focal conic texture was 30 V (Figure 3b), which was much smaller than that of the PVA–PVA cell. To further clearly show the switching difference, we plot the transmittance of both the cells at the wavelength of 500 nm (extracted from the transmittance grooves) versus the applied voltage. As shown in Figure 3c, it is obvious that the switching voltage of the PVA–PVK cell from cholesteric texture to focal conic texture is 15 V, smaller than that of the PVA–PVA cell.

The mechanism behind this is that the PVK layer can generate hole–electron pairs under exposure to UV light, which results in the reduction in the built-in electric field in the PVA–PVK cell [32]. Specifically, when an external DC voltage is applied, the built-in electric field $\left(E_{b}\right)$ is generated from the impurity ions of the LCs. The $E_{b}$ is opposite to the external field ($E_{f}$), and, thus, limits the external field, as shown in Figure 4a. The effective electric field ($E_{eff}$) in the cell as derived from the external field (E*_f_*) can be written as
(2)$$E_{eff}=E_{f}-E_{b}$$

In the PVA–PVK cell, the generated hole–electron pairs in PVK layer in the presence of UV light can neutralize the impurity electrons in the LC–PVK junction (Figure 4b) [32]. Here, the UV light is from the tungsten lamp during the test. Therefore, the built-in electric field ${E_{b}}^{\prime }$ in the PVA–PVK cell is smaller than that ($E_{b}$) in the PVA–PVA cell. For the same $E_{eff}$, it needs a smaller $E_{f}$ to drive the LC in the PVA–PVK cell to realign from cholesteric texture to focal conic texture. Appendix A visually shows the effect of the PVK layer on reducing the operation voltage of the CLC device with the assist of UV light. With holding applied voltage of 30 V, the PVA–PVA cell keeps a colored transparent state due to the cholesteric texture, regardless of UV irradiation or not. For the PVA–PVK cell, under the same applied voltage, the cell turns to opaque from the colored transparent state under UV irradiation, which means UV light can help to switch the cell from cholesteric texture to focal conic texture.

Since the PVK alignment layer can help to reduce the $E_{b}$ in LC cells, we further investigated the effect of the PVK layer on reducing the operational voltage of the infrared reflector based on polymer stabilized cholesteric LC (PSCLC). Mixture 2 was injected into the anti-parallel rubbed PVA–PVA cell and PVA–PVK cell, respectively. The cells were then exposed to UV light for polymerization. We then measured the transmittance of the cells under an applied electric field through a spectrometer. Figure 5 illustrates the electrically tunable bandwidth broadening of the cells. The transmittance spectra exhibited a reflection band centered at 1100 nm with an initial bandwidth of around 120 nm for both cells. In the visible region, the transmittance is above 85%. The inset photos in Figure 5a,b show the POM images and prototypes of the cells in the initial state. When a DC voltage was applied, the reflection band is extending simultaneously towards both the red and blue sides compared to the original reflection notch wavelength (Figure 5a,b), resulting in bandwidth broadening. This can be explained by a non-uniform pitch distribution due to the ion-dragging effect in PSCLC systems under the applied electric field [20,27]. Moreover, on application of the same voltage, for the PVA–PVK cell, the bandwidth broadening is larger than that of the PVA–PVA cell (Figure 5c). For example, at 60 V, the reflection bandwidth of the PVA–PVK cell is 821 nm while it is 647 nm for the PVA–PVA cell. In other words, the PVK alignment layer can help to achieve the same bandwidth broadening at a lower operation voltage. This is also attributed to the reduction of $E_{b}$ in the PVA–PVK cell due to the LC–PVK junction (Appendix A).

Since the UV irradiation plays an important role in generating hole–electron pairs in the PVK layer, resulting in the reduction of $E_{b}$, we further changed the UV irradiation condition by increasing the UV dose to see if it could influence the bandwidth broadening of the IR reflector. With holding 20 V voltage on the IR reflector sample, we exposed the sample to an extra UV irradiation for different UV doses, and measured the transmittance of the sample (Appendix A). Figure 5d exhibits the bandwidth of the sample under different UV doses. Without the extra UV irradiation, the bandwidth was 269 nm under the applied voltage of 20 V, which had already been broadened compared with the PVA–PVA cell under the same applied voltage, which was 215 nm (Figure 5c). Under the extra UV irradiation with dose of 0.5 J/cm^2^, the bandwidth extended to 305 nm. When the UV dose was increased to 2 J/cm^2^, the bandwidth increased to 471 nm, which was close to the bandwidth of the cell on application of 40 V when there was no extra UV irradiation. Obviously, with the assist of UV irradiation, the IR reflector can achieve an enhanced bandwidth broadening at a lower voltage, and the bandwidth is increasing as the illuminated UV dose is larger.

## 4. Conclusions

In summary, we have reported in this work on the preparation of a CLC and PSCLC device in which PVK was used as the alignment layer. The POM images and calculated order parameter of a nematic LC cell with the PVA–PVK alignment layer indicate that the combination of the PVA–PVK can induce a homogeneous alignment of LCs. Due to its semi-conductive property, there are hole–electron pairs generated in the PVK layer in the presence of UV irradiation, which neutralize the impurity electrons in the LC–PVK junction, resulting in the reduction of the built-in electric field in the LC device. As a result, the operational voltage of the CLC device switching from cholesteric texture to focal conic texture decreases from 45 V to 30 V. For the PSCLC-based IR reflectors with the PVK alignment layer, at the same applied electric field, the reflection bandwidth is enhanced from 647 to 821 nm in the IR region. Our results demonstrate a feasible way to fabricate PSCLC-based infrared reflectors with low operational voltages and enhanced reflection bandwidth, making it attractive for saving energy as a smart window.

## Data Availability

The authors confirm that the data supporting the findings of this study are available within the article.

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
