# Peer review of "Enhanced Bandwidth Broadening of Infrared Reflector Based on Polymer Stabilized Cholesteric Liquid Crystals with Poly(N-vinylcarbazole) Used as Alignment Layer"

_polymers, 2021, doi:10.3390/polym13142238_

Round 1

Reviewer 1 Report

Review of the article “Enhanced bandwidth broadening of infrared reflector based on polymer stabilized cholesteric liquid crystals with Poly(N-vinylcarbazole) used as alignment layer” by authors Limin Zhang, Qiumei Nie, Xiao-Fang Jiang, Wei Zhao, Lingling Shui, Xiaowen Hu, and Guofu Zhou.

In this work, the authors used a semiconductor material poly (N-vinylcarbazole) (PVK) as an aligning coating for the fabrication of LCD devices. Under ultraviolet irradiation, hole-electron pairs are generated in the PVK layer, which neutralizes impurity electrons in the LC - PVK junction, which leads to a decrease in the built-in electric field in the LCD device and, as a consequence, to a decrease in the control voltage threshold. The authors investigated the influence of the aligning layer of PVK on the switching of CLCs with positive anisotropy and on the enhancement bandwidth of PSCLC broadening with negative anisotropy. In the case of CLCs with positive anisotropy, they obtained a decrease in the control voltage, which switches the Granjan texture of the CLC into a focal-conical one from 45 V to 30 V. In the case of PSCLC, they obtained enhancement in the bandwidth of the selective reflection from 647 nm to 831 nm.

However, there are some inconsistencies in the article:

  • Missing letter "r" in word "through" on line 208.
  • In the text of the article, I found a mention of Figure S1 on line 195, Figure S2 on line 221 and Figure S3 on line 233. However, I did not find Figures with such designations. Judging by the context, Figures S1 and S2 correspond to Figure 4b, and Figure S3 corresponds to Figure 5d.
  • On line 130 the authors write that “However, for the cell with PVK-PVK alignment layer, no matter how to rotate the cell under the crossed polarizer, the dark field and bright field cross each other in all the images (Figure1c), which indicating an unordered orientation”. However, this is not quite true; the alternation of the bright and dark fields in this case also takes place. It is another matter that the fields are not very bright and not very dark and, in addition, contain many lines of disinclinations. Therefore, it cannot be said that in the case of a PVK-PVK cell, a disordered orientation arises. Moreover, the authors measured the order parameter in this case, which is 0.37. And also in the resume they write on line 245 “a PVK layer can induce a homogeneous alignment of LCs”.

Despite minor inconsistencies, which I believe will be corrected, the article can be published in the journal "Crystals", since the results obtained are not in doubt and are original.

Reviewer 2 Report

This work studies the role played by the semi-conductive material (PVK) used as alignment agent for liquid crystals (LCs). The work uses cells treated with the alignment agents polyvinyl alcohol (PVA), PVK or hybrid cells which are filled with CLC mixtures with photonic band gaps (PBG) in the infrared region. Under the application of an electric field an important broadening of the photonic band gap (PBG) is observed. The PVK layer reduces the electric field required for PBG widening respect to cells with only PVA alignment agent.

The manuscript is interesting and presents new results. The description of the experiments and the interpretation of the results are clear and conclusive. Therefore I think that the work is suitable to be considered for publication in Polymers. Only some minor points should be addressed.

-An interesting work, by Wang Hu et al, which describes the broadening of the PBG of a polymer stabilized CLC by means of electric fields, should be added in the references and perhaps commented in the introduction: "Electrically Controllable Selective Reflection of Chiral Nematic Liquid Crystal/Chiral Ionic Liquid Composites" Adv. Mater. 2010, 22, 468–472.

- Reference 30 is incomplete.

- The PBG broadening in polymer stabilized CLCs under electric field is assumed to be due to the polymer network motion due to the migration of ionic impurities.  However, in this work the UV irradiation produces electron-hole pairs in the CLC-PVK junction that neutralize the impurity electrons. In principle this effect would seem to be detrimental for the bandwidth broadening enhancement. Could you comment this point?
